# Sustainable Recovery of Anthocyanins and Other Polyphenols from Red Cabbage Byproducts

**DOI:** 10.3390/foods12224157

**Published:** 2023-11-17

**Authors:** Rusăndica Stoica, Mihaela Ganciarov, Diana Constantinescu-Aruxandei, Luiza Capră, Ioana-Raluca Șuică-Bunghez, Raluca-Mădălina Senin, Georgiana Diana Pricope, Georgeta-Ramona Ivan, Costin Călin, Florin Oancea

**Affiliations:** 1Analysis and Bioresources Departments, National Institute for Research & Development in Chemistry and Petrochemistry—ICECHIM, Splaiul Independentei No. 202, Sector 6, 060021 Bucharest, Romania; rusandica.stoica@icechim.ro (R.S.); mihaela.ganciarov@icechim.ro (M.G.); diana.constantinescu@icechim.ro (D.C.-A.); luiza.capra@icechim.ro (L.C.); raluca.bunghez@icechim.ro (I.-R.Ș.-B.); raluca.senin@icechim.ro (R.-M.S.); diana.pricope@icechim.ro (G.D.P.); georgeta.ivan@icechim.ro (G.-R.I.); 2Iprint3D Design & Consulting Srl, Str. George Enescu No.5, Sector 3, 030167 Bucharest, Romania; costincalin1@gmail.com; 3Faculty of Biotechnologies, University of Agronomic Sciences and Veterinary Medicine of Bucharest, Mărăști Blv., No. 59, Sector 1, 011464 Bucharest, Romania

**Keywords:** red cabbage byproducts, apple vinegar, total monomeric anthocyanin pigment content, total polyphenol content, storage stability, food color additives, organic fertilizers

## Abstract

The objective of this work was to develop a sustainable process for the extraction of anthocyanins from red cabbage byproducts using, for the first time, apple vinegar in extractant composition. Our results showed that the mixture 50% (*v*/*v*) ethanol–water, acidified with apple vinegar, used in the proportion of 25 g of red cabbage by-products per 100 mL of solvent, was the best solvent for the preparation of an anthocyanin extract with good stability for food applications. The chemical characterization of this extract was performed by FTIR, UV-VIS, HPLC-DAD, and ICP-OES. The stability was evaluated by determining the dynamics of the total polyphenol content (TPC) and the total monomeric anthocyanin pigment content (TAC) during storage. On the basis of the statistical method for analysis of variance (ANOVA), the standard deviation between subsamples and the repeatability standard deviation were determined. The detection limit of the stability test of TPC was 3.68 mg GAE/100 g DW and that of TAC was 0.79 mg Cyd-3-Glu/100 g DW. The red cabbage extract has high TPC and TAC, good stability, and significant application potential. The extracted residues, depleted of anthocyanins and polyphenols with potential allelopathic risks, fulfill the requirements for a fertilizing product and could be used for soil treatment.

## 1. Introduction

Color represents an essential sensory attribute for food, which is directly related to customer perception of quality and the success of a product on the market [1]. Consumer acceptance of synthetic dyes used as food additives is constantly decreasing because of concerns related to health risks and environmental impacts [2]. The food industry tends to use natural pigments extracted from microorganisms [3] and plant by-products [4,5].

Natural pigments extracted from plant by-products represent various organic compounds with unique physicochemical properties [6]. They include a diverse class of chemicals, such as chlorophyll, carotenes, anthocyanins, flavonoids, and betalains, which selectively absorb light at specific wavelengths while reflecting it at other wavelengths [5]. The pigments from natural sources cover various color palettes; anthocyanins from grapes, berries, and red cabbage are red, purple, and blue, chlorophylls are green, and carotenes from carrots, corn, and tomatoes have colors from yellow to orange-red [7]. One of the main advantages of natural pigments is their bioactivity, which makes them multipurpose food additives. Antioxidant activity is common to natural pigments: carotenoids and chlorophyll [8], flavonoids [9], anthocyanins [10], and betalains [11]. The recovery of natural pigments from the by-products of edible plants valorizes significant quantities of side streams generated by plant-based food value chains [6,12].

Among natural pigments, anthocyanins are the largest water-soluble group [13]. These pigments exist in many structural forms, both simple and complex, and are governed by certain physicochemical phenomena that have profound effects on their colors and stabilities [14]. Anthocyanin pigments are important for food quality because they contribute to color and appearance. The health benefits of anthocyanins include, besides antioxidant effects, the following activities: prebiotic [15], cancer prevention [16], eye health promotion [17], anti-hypertensive [18], anti-diabetic [19], and control of obesity [20] and of metabolic syndrome [21]. Owing to their health benefits, the anthocyanin pigment content can be a helpful criterion in quality control and the purchase specifications of fruit juices, nutraceuticals, and natural colorants [22].

Red cabbage (*Brassica oleracea* var. *capitata F. rubra*) is a good source of carbohydrates (0.7–5.3%), fibers (1.0–3.6%), proteins (0.8–2.0%) and minerals (0.3–0.7%), and is usually consumed as a fresh vegetable [23]. The main coloring pigments of red cabbage are anthocyanins, with red cabbage having the highest anthocyanin content among leafy vegetables [24]. The red cabbage products contain twenty different nonacylated and acylated anthocyanins with the main structure of cyanidin-3-diglucoside-5-glucoside [25].

As there are no specific data available on red cabbage, the global data on cabbage show that approximately 72 million tons are produced worldwide per year. China and India together produce 59% of the world total production. Europe produced 9.6 million tons of cabbage in 2020, and Romania produced 600,550 tons, being the third largest producer in Europe after Germany and Poland according to the Food and Agriculture Organization of the United Nations (FAO). The percentage of waste generated from processing can have values between 12 and 63% depending on the type of material and the transformation process. FAO has estimated that 30% of the total food produced each year is lost or wasted. This can lead to significant environmental impacts, as well as economic and social effects [26].

Considering these aspects, our work aimed to develop a sustainable and affordable process for the recovery of anthocyanins from red cabbage waste using an ethanolic aqueous extractant that includes apple vinegar. Such an extractant composition is reported for the first time to the best of our knowledge. The study also brings insight into the effect of vinegar on the recovery of polyphenolic compounds in different extracting mixtures and the contribution of extractants before the extraction process. The quality of the extract obtained under optimal conditions was investigated in order to determine the physicochemical properties by UV-Vis, FTIR, HPLC-DAD, and ICP-OES. In addition, the sustainable process that we developed in this work aims to demonstrate the possibility of valorizing and closing the loop of red cabbage value chains in a biomimetic manner, i.e., returning nutrients to the soil. Cabbage is a crop that requires intense fertilization for optimum yield [27]. We evaluated the suitability of the anthocyanin (and polyphenols) from extracted red cabbage biomass for its use as an organic fertilizing product to return nutrients to the soil. This would ensure a zero-waste process.

## 2. Materials and Methods

### 2.1. Chemicals

Gallic acid, 6-hydroxy-2,5,7,8-tetramethylchroman-2-carboxylic acid (Trolox), 1,1-diphenyl-2-picryl-hydrazyl (DPPH), catechin, epicatechin, quercetin 3-rutinoside, and trifluoroacetic acid were purchased from Sigma-Aldrich (St. Louis, MO, USA); kaempferol was supplied by Cayman Chemical (Ann Arbor, MI, USA), Folin–Ciocalteu USP 224 was from CPA Chem (Stara Zagora, Bulgaria); methanol and HPLC Plus Gradient were from Carlo Erba Reagents (Val de Reuil, France); ethanol from ChimReactiv (Bucharest, Romania) and LiChrosolv from Merck (Darmstadt, Germany); pH buffers (pH 4.00, pH 7.00, pH 10.00) were purchased from Scharlab (Barcelona, Spain); and apple vinegar with total acidity 10% expressed in acetic acid and a pH = 2.31 at 20 °C was from Vitaplant (Târgu-Mureș, Romania). The certified reference materials for elemental analysis (sulfanilamide, 2,5-bis(5-*tert*-butyl-benzoxazol-2-yl)thiophene (BBOT) and methionine) were obtained from Thermo Fischer Scientific (Waltham, MA, USA). For the ICP-OES standard, Certipur solutions, with a concentration of 1000 or 100 mg/L, from Merck (Darmstadt, Germany), were used. Ultrapure water produced by the Milli-Q Integral System from Merck (Darmstadt, Germany) with a resistivity of 18.2 MΩ/cm was used for the preparation of working solutions and samples. The purge gas for ICP-OES was Argon 5.0 of 99.999% purity, purchased from Siad (Călărași, Romania). All working standard solutions were freshly prepared before use. All reagents used in the experiments were of analytical grade.

### 2.2. The Extraction Process

Fresh red cabbage by-products from food processing, consisting of damaged leaves mixed with stubs, were rinsed with distilled water and cut into small pieces. The extraction process is characterized by the following steps: homogenization of the red cabbage by-products (25 g) in a blender, mixing with 100 mL extraction solvent mixture, followed by ultrasound-assisted extraction (UAE, Fritsch ultrasound bath, Laborette, sound power 2 × 240 W/period) of the plant material for 20 min at 25 ± 5 °C after flushing the flask with nitrogen. The extraction solvents were selected based on the previous experience of the authors [28,29], and the data from the literature [30]. The following aspects were considered: the use of non-toxic solvents, the limitation of organic substances used, the reduction of the extraction time and the generation of products with added value. Anthocyanins are stable in acid medium, so the extraction was carried out in the presence of apple vinegar and as a solvent. In addition, acetic acid is known as a chemical preservative and also show a good antibacterial activity [31]. Therefore, the solvents used for extractions were bidistiled water and 30%, 50%, 70% (*v/v*) ethanol solutions. The concentration of vinegar in the solvent mixture was 10% (*v*/*v*), and the pH values of the extractants were determined. Each extract was centrifuged using a Universal 320 R Hettich centrifuge (Tuttlingen, Germany) at 1140 rcf and 4 °C for 10 min., and the supernatant was filtered. The extraction steps were repeated three times. The resulting total volume of extract was collected and concentrated on a rotary evaporator Antrieb-W-Mikro (Heidolph Instruments, Schwabach, Germany) at 35 °C under vacuum until a residual volumetric content of 20% from the initial volume, which resulted in a 5× concentrated extract. The mass of resulted extract was weighed on the analytical balance XS 204 (Mettler Toledo, UK). The extracts were characterized in terms of pH, color density, degradation index, total polyphenol content (TPC), expressed as gallic acid equivalent (GAE), and total monomeric anthocyanin content (Cyd-3-Glu) via the pH differential method.

The red cabbage extract obtained under optimal conditions was characterized by FTIR, antioxidant activity, ICP-OES, short-term stability study (TPC and TAC), and HPLC-DAD as described below. The residue after the three extraction steps was characterized by elemental analysis.

### 2.3. Analytical Methods

#### 2.3.1. FTIR Spectroscopy

FTIR analysis of the best red cabbage extract and polyphenol standards was performed in the attenuated total reflection (ATR) mode, with a Spectrum GX spectrometer (Perkin Elmer, Waltham, MA, USA) in the range 4000–1600 cm^−1^, with 32 scans and a resolution of 4 cm^−1^. Spectrum software v. 5.3.1 was used for spectrum processing and interpretation.

#### 2.3.2. Determination of the Total Phenolic Content (TPC)

The Folin–Ciocalteu assay [32] was applied with some modifications to determine the TPC of the samples. The optical densities of the reaction mixtures prepared according to [33] were measured after 60 min against water in 10 mm path length cells using a Cintra 202 spectrophotometer (GBC Scientific Instruments, Keysborough, VIC, Australia) set at 765 nm. The total phenolic content of red cabbage extracts was expressed as the gallic acid equivalent from the calibration curve (y = 0.011x + 0.0205; R^2^ = 0.9997) per 100 g of raw sample dry weight (DW). The total phenolic content of the solvents was determined before the extraction step in order to observe their contribution to the extraction process.

#### 2.3.3. Quantification of Total Anthocyanins

The total monomeric anthocyanin content (TAC) was determined using the differential method [34]. The absorbances of the samples were measured at λ_vis-max_ (wavelength of maximum absorbance) and at 700 nm and the results are expressed as milligrams of cyanidin-3-glucoside (molar extinction coefficient of 26,900 L × mol^−1^ × cm^−1^ and molecular weight of 449.2 g/mol) equivalents per 100 g raw sample DW [34].

#### 2.3.4. Determination of Total Flavonoids

The total flavonoid content was measured by the method developed by Zhishen [35]. The absorbance values at 510 nm compared to a blank were recorded, both for the standard catechin solutions and for the analyzed samples. Total flavonoids of the colored extracts were expressed as mg catechin equivalents per 100 g DW.

#### 2.3.5. Determination of Antioxidant Activity (AOA)

The method used for AOA is based on the decolorization of the stable radical 2,2-diphenyl-picrylhydrazyl (DPPH) colored red-purple with absorption at 517 nm by substances with an antioxidant activity and expressed as percentage of DPPH scavenging and as Trolox equivalent (TEAC) [36,37]. Briefly, a stock solution of 0.3 mM DPPH was prepared in ethanol. The three different reaction systems prepared according to [28] were incubated in the dark for 1 h, and the absorbance was read at 517 nm. A calibration curve was made in the Trolox concentration interval of 0.01–0.05 mg × mL^−1^.

#### 2.3.6. Determination of Color Density

Each sample was diluted with water, and the color density was determined using Equation (1) [38].
Color density = [(A_420 nm_ − A_700 nm_) + (A_λ vis-max_ − A_700 nm_)] × DF(1)

#### 2.3.7. Degradation Index

This was calculated from the absorbances of water-diluted red cabbage solutions [39].
Degradation index = [(A_420 nm_ − A_700 nm_)/(A_λ vis-max_ − A_700 nm_)] × DF(2)

#### 2.3.8. ICP-OES Analysis

The Optima 2100 DV ICP-OES System (Perkin Elmer, Waltham, MA, USA) was used with a dual-view optical system—axial and radial views of the plasma in a single work sequence, which works with an independent transistorized radio frequency generator with a frequency of 40 MHz. The nebulization system is equipped with a PEEK Mira Mist nebulizer coupled with a baffled cyclonic spray chamber. The spectrometer is made up of an optical module that includes an Echelle monochromator with a two-dimensional charged coupled device) detector, with a spectral range of 165–800 nm. For the determination of the elements in the red cabbage extract, approximately three grams of sample were weighed and subsequently mineralized with HNO_3_ 65%, HCl 37%, and H_2_O_2_ 30% (6:2:2) on the sand bath and then adjusted quantitatively to a 25 mL volumetric flask with ultrapure water, obtaining a clear solution. For the red cabbage residue analysis, the same procedure was applied with 0.5 g sample and a 50 mL solution after mineralization. The samples were analyzed in triplicate. In parallel with the samples, a blank sample was prepared under the same conditions. WinLab32 software v.5.5.0.0714 was used for spectrum processing and interpretation.

#### 2.3.9. The Stability Study

The stability of the red cabbage extract was tested by measuring the total polyphenol and monomeric anthocyanin contents at the beginning of the storage period and after 7, 14, and 21 days. The sample was divided into four subsamples, of which one (E1) was analyzed immediately and the others were kept in the temperature range between 2 °C and 4 °C until analysis. The stability assessment was carried out by determining the difference between the subsample averages, analyzed at the set time, and the E1 subsample average. Three tests were performed on each sample under repeatability conditions.

The relationship between the sources of variation, as presented in Table 1, which outlines the sums of squares and degrees of freedom, was used for statistical analysis. The repeatability standard deviations (sr), mean squares between groups MSE_between_, and mean squares within groups MSE_within_ were calculated. On the basis of MS_within_ [40,41], the s_measured_ (s_meas_) and the detection limit of the stability (LOD) test were calculated, as detailed in Appendix A.

#### 2.3.10. HPLC-DAD Analysis

The analysis of flavonoids was performed using the HPLC 1100 system with DAD detector Agilent (Santa Clara, CA, USA). The separation of catechin, epicatechin, quercetin 3-rutinoside (Que-rut), and kaempferol was performed on a Kromasil 100-5C18 column (length × inner diameter: 150 mm × 4.6 mm, 5 μm particle size, pore size 100 Å) with the method described by Stoica et al. [42]. The mobile phase was a mixture of two eluents: A, water with 0.03% trifluoroacetic acid (TFA), and eluent B, methanol. The flow rate was 1 mL/min and the injection volume was 10 μL. Simultaneous monitoring was set at 280 nm for catechin, epicatechin, and kaempferol, and 350 nm for quercetin-3-rutinoside. Individual stock standard solutions of 100 mg × L^−1^ were prepared in methanol, and the calibration standards were obtained by appropriate dilution from the stock solution. All working standard solutions and studied samples were filtered by a 0.45 μm pore size syringe filter.

#### 2.3.11. Elemental Analysis of the Red Cabbage Extract Residue

The CHNS FlashSmart™ Elemental Analyzer (Thermo Fischer, Waltham, MA, USA) was used to determine the elemental compositions of the extracted red cabbage dried at 40 °C. The results were calculated from the calibration curve obtained using certified reference materials of sulfanilamide and 2,5-bis(5-*tert*-butyl-benzoxazol-2-yl)thiophene (BBOT). The correlation coefficient obtained was higher than 0.999. The calibration curve was verified using a methionine certified reference standard.

#### 2.3.12. Statistical Analysis

Data were statistically analyzed using Microsoft Excel software version 2310. Significant differences between groups were determined using one-way analysis of variance (ANOVA). Furthermore, IBM SPSS Statistics software version 26.0.0.0 was used for Tukey’s test in order to find out the significance between groups (*p* ≤ 0.05), and all the data were presented as mean ± standard error having triplicate analysis (n = 3).

## 3. Results and Discussion

Various extraction systems have been reported for the recovery of anthocyanins from red cabbage, such as acetone or mixtures of acetone [43], methanol [44], water/ethanol [30], and many organic or inorganic acids [45]. Our study focused on the sequential use of red cabbage by-products. As the toxicity/restrictions related to the use of some solvents (acetone, methanol) are well known, we used H_2_O or ethanol acidified aqueous solutions for the anthocyanin extraction. Because it was previously observed that acylated anthocyanins are degraded in solvents containing mineral acids [46], we used apple vinegar (10% *v*/*v*) to extract anthocyanins and other polyphenols. Because anthocyanins are sensitive to heat [47], the temperature applied during the extraction was between 25 and 30 °C.

The choice of acidified water and acidified ethanol aqueous solutions for anthocyanin recovery from red cabbage by-products is based on the polar character of the anthocyanin molecule. Additionally, extraction in the presence of weak organic acids is recommended to minimize the decomposition of pigments [43]. To the best of our knowledge, edible apple vinegar has never been reported for anthocyanin extraction from red cabbage. Therefore, for extraction, we used H_2_O or ethanol aqueous solutions acidified with apple vinegar diluted to a concentration of 1% (*v*/*v*) acetic acid (AA). A relatively low concentration of acetic acid was chosen for this study because some authors observed that 5% acetic acid appeared more damaging to the cell than the McIlvaine buffer solution pH 3.0 (citrate–phosphate buffer) [30].

### 3.1. Composition of the Solvent Systems Used for the Extraction

It is known that the amount of organic acids in natural vinegar is mainly influenced by the raw materials and the fermentation method, which affect acidity and pH. Acidity is associated with the concentration of chemical species that can release protons in an aqueous solution. The pH of a solution is related to the concentration of H^+^ ions in the solution at equilibrium [48]. It can be observed from the solvent analysis results before the extraction process (Table 2) that the extraction solvent with glacial acetic acid has a lower pH than the solvents based on apple vinegar. The solvent systems with apple vinegar had higher TPC values than the TPC with acetic acid, the value of which can be considered an interference due to the presence of acetic acid.

Tukey’s test was performed at 95% confidence level to compare the mean of groups. Table 2 shows that the pH of extractants with apple vinegar share two letters each ab, bc and cd which reflect “overlap” between the sets of groups. The difference can be considered as marginally statistically significant. The pH of the solvent systems without apple vinegar that do not share a letter have a mean difference that is statistically significant. For TPC, the result of the test showed that there was statistically significant difference between TPC determined in solvent systems before the extraction process.

### 3.2. Effect of the Solvent System on Polyphenol and Anthocyanin Recovery

The extraction capacities of several solvents studied are shown in Table 3. The extracts were analyzed for total monomeric anthocyanin pigment (TAC), total polyphenol content (TPC), color density, pH, and degradation index.

The total polyphenol content of the red cabbage extracts was between 518.20 and 2272.07 mg GAE/100 g DW, depending on the extraction solvent. Recent studies reported TPC values of 1626.2 mg GAE/100 g in an EtOH 70% extractant acidified with 0.1% acetic acid [49]. The results obtained for TAC ranging from 128.57 mg in water to 374.70 mg cyanidin-3-glucoside (Cyd-3-Glu)/100 g in EtOH 50%-AA1% are in agreement with data reported by other authors of 336.7–626.3 mg Cyd-3-Glu/100 g extracted with 70% methanol acidified with HCl [50].In the acidified extractants, the TAC and TPC were about 3–4-fold higher and the degradation index decreased 4–5 times than in water, which suggests a significant stabilization of the extracted biomolecules that can be attributed to the acetic acid medium. The 50% (*v*/*v*) ethanol acidified aqueous solution gave the highest content of total monomeric anthocyanins, total polyphenol content, and color density. The results obtained are in good correlation with those reported by other authors who observed an increase in the diffusion coefficients of anthocyanins in aqueous ethanol solutions with concentrations between 39% and 67% (*v*/*v*). At higher concentrations of 67–95% ethanol, the diffusion coefficient decreases [51]. This behavior is probably related to the changes in solvent polarity and partially in viscosity; at low ethanol concentrations, the polarity and viscosity are too high for efficient extraction, and at high ethanol concentrations, the polarity decreases too much, limiting the extraction. A small contribution of 0.8–1.3% from TPC obtained during the extraction process can be assigned to the solvent system based on apple vinegar. Natural vinegars are known to contain various bioactive compounds such as sugars, organic acids, polyphenols, vitamins, amino acids, melanoidins, and tetramethylpyrazine with synergistic action [52].

The color density as a measure of color strength had values between 1.05 and 4.12, with the highest values being obtained for the ethanolic extracts. The degradation index in water (1.05) was approximately 4 times higher than in the other extraction systems (0.21–0.27) presented in Table 3, which shows that the used acid medium stabilizes the biomolecules and is suitable for the extraction process. The results obtained are in agreement with those reported by other authors [39], with values between 0.24 and 0.32 for the 48% ethanolic extracts of red cabbage.

Moreover, Tukey’s multiple comparison test was applied to check which extractant means were significantly varying from each other. The statistical analysis for pH showed that only in one case (water), there was a statistically significant difference compared to the rest of the extractants (Table 3). Regarding TAC results, the test did not detect a significant statistical difference between the solvent system with 50% and the one with 70% ethanol. Similar observation for TPC and color density between extractants with 30 and 70% ethanol. For degradation index, the extractants with 50% and 70% ethanol both share letter ab which reflects “overlap” between the sets of groups. However, it was observed that the solvent systems that do not share a letter have a mean difference that is statistically significant.

Because the 50% (*v*/*v*) ethanol acidified aqueous solution was the most effective solvent, it was chosen for pigment characterization and stability testing.

The chromatographic separation of the 50% (*v*/*v*) ethanolic red cabbage extract with and without apple vinegar by HPLC-DAD is presented in Figure 1. The identification of flavanols (catechin, and epicatechin) and flavonols (quercetin 3-rutinoside, and kaempferol) compounds was based on the retention time determined in the sample compared with reference materials.

The presence of epicatechin between 0.15 mg/100 g [49] and 218.93 mg/100 g DW [50] was previously reported, depending on the production process or vegetative form of the plant. Our data fit within this range. The results of our research (Table 4) showed that the quercetin 3-rutinoside content of red cabbage extract obtained in the presence of apple vinegar was approximately 3-fold higher than that of extract without vinegar.

From the literature data, the authors observed that the concentration of quercetin 3-rutinoside varied from 4.88 ± 0.02 mg/100 g in mature vegetables to 33.27 ± 2.06 mg/100 g DW in young shoots from an extract obtained with 70% methanol acidified with HCl [53]. We obtained higher concentration (62.96 mg/100 g DW) than previously reported by applying a solvent system of EtOH 50% with apple vinegar.

The HPLC-DAD profile (Figure 2) shows that the content of the analyzed flavonoids in the extraction solvent with apple vinegar has no direct contribution to the determined compounds in the red cabbage extract being below the quantification limits. However, the determined flavonoid of 62.92 mg/100 g DW from the red cabbage acidified extract can be attributed to the vinegar composition effect on the extraction process.

On the other side, the levels of TFC in red cabbage determined based on the calibration curve were 367.73 ± 6.30 mg/100 g DW and 205.30 ± 10.33 mg/100 g DW, respectively. From the literature, the TFC concentration had a value of 209.9 ± 0.03 mg/100 g (DW) extracted with 70% ethanol [54]. Based on the results obtained, it could be concluded that the extraction of polyphenols in the presence of apple vinegar represents a good option that can influence the recovery of bioactive compounds for sequential use of red cabbage waste by-products.

### 3.3. Extract Characterization

The red cabbage extract obtained was characterized by Fourier transform infrared spectroscopy, UV-VIS molecular absorption spectrometry (total content of polyphenols, anthocyanins and antioxidant activity) and by inductively coupled plasma optical emission spectrometry (ICP-OES).

#### 3.3.1. Analysis by FTIR Spectroscopy

The FTIR spectrum of the red cabbage extract (Figure 3) shows a band at 3284 cm^−1^ associated with the stretching vibration of O–H bonds. At 2935 cm^−1^ a characteristic band of the C–H stretching vibration present in aliphatic chains was observed, and at 1726 cm^−1^ and 1663 cm^−1^ the characteristic stretching vibration of the C=O flavonoids was observed, confirming the presence of aromatic compounds in the red cabbage extract. Another band was observed at 1516 cm^−1^ corresponding to the axial deformation of the C=C bond in aromatic rings [55].

The band at 1414 cm^−1^ can be assigned to the C–O groups, which display the angular deformation of phenols [53]. The spectrum also showed a band at 1024 cm^−1^, resulted from stretching vibration of the anhydroglucose ring of O–C in flavonoid compounds [56]. Other bands were observed between 1000 cm^−1^ and 600 cm^−1^; corresponding to aromatic C–H [55]. According to the FTIR spectra of red cabbage extract and compared with standards of phenolic acids (gallic acid) and flavonoids (catechin and quercetin 3-rutinoside) and with literature data [57,58], it was found that the samples present both bands specific to carboxylic acids (spectral area 1726–1650 cm^−1^) and bands specific to flavonoids (spectral area 1650–1400 cm^−1^ and the spectral area 1000–600 cm^−1^). Because anthocyanins are composed of hydroxyl groups, benzene rings, and oxygen-containing heterocycles, the characteristic skeleton vibration peaks of anthocyanins are mainly in the regions of 1140−1000 cm^−1^ and 930−700 cm^−1^ [56].

#### 3.3.2. Antioxidant Activity (AOA)

The 50% (*v*/*v*) ethanol acidified aqueous solutions exhibited high antioxidant activity as measured by the DPPH assays with a value of 77.69 ± 2.20%. The AOA determined by the DPPH method [59] as the Trolox equivalent (TEAC) was 0.93 ± 0.04 mg/g based on the equation of the calibration curve (y = 1653.330x + 12.989). The determined antioxidant activities are in agreement with other reported results for radical scavenging activity (RSA) of 70.92% and TEAC 1.94 mM Trolox equivalent [60]. The antioxidant activity of the 50% (*v*/*v*) ethanol aqueous solutions without vinegar was 0.68 ± 0.03 mg/g TEAC. The AA of the best extraction solvent system without red cabbage extract was below 0.10 mg/g TEAC value, which represents the limit of quantification of the method.

#### 3.3.3. Analysis by ICP-OES

The analysis of the extracts to determine the content of Na, K, P, Ca, Mg, Al, Zn, Fe, Mn, Ni, Cr, Pb, Cd, and Co was carried out by maintaining the same conditions as when drawing the calibration curve. For the analytical determination of the elements, calibration curves were drawn using the standards obtained by dilution from Certipur standard solutions of 1000 and 100 mg/L in the working range specified in Table 5. The concentrations of the elements in the red cabbage extract (EtOH50%-AA1%) are presented in the same table. The content of essential mineral elements found in the red cabbage extract is mainly represented by potassium (K), calcium (Ca), magnesium (Mg), and phosphorus (P). Small quantities of Zn, Fe, and Mn were also detected. In the analyzed extract, the contaminants Ni, Cr, Cd, Pb, and Co had values lower than the quantification limits.

The results obtained are comparable with the literature data for K (2060 mg/Kg), P (420 mg/Kg), [61], Na (46.5–55.0 mg/Kg), and Mg (18.0–19.5 mg/Kg) [62]. The addition of metal ions to some matrices for color stabilization could be a viable alternative for dye production, especially since some metal ions pose no health hazard [63]. The stabilization of the color of anthocyanins could be due to the ability of metals to prevent the oxidation of the quinoid blue form. Al, Fe, Cu, Sn, Mg, and Mo were among the first elements to demonstrate the ability to form metal complexes with anthocyanin molecules [63]. The presence of heavy-metal contaminants at values below the limit of detection indicates a very low content, which is an important observation in terms of the environment and safety.

#### 3.3.4. The Short-Term Stability Study

One of the main drawbacks of anthocyanin extracts from red cabbage is their low stability [64]. Therefore, our extraction process also targeted the extraction of other antioxidant polyphenols from the red cabbage by-products. We investigated the short-term storage stability in the temperature range between 2 °C and 4 °C of the red cabbage extract resulted under optimum condition. The short-term stability was also studied because it represents a key parameter for various applications, including further formulation by encapsulation [65] or as pigments in the ink for additive manufacturing, i.e., 3D/4D printing [66], which represent the subject of another paper.

The stability of the obtained colored extracts was evaluated in terms of the total content of polyphenols, expressed in mg/100 g, as gallic acid equivalent (GAE) and total monomeric anthocyanin pigment content (TAC), expressed in mg/100g cyanidin-3-glucoside (Cyd-3-Glu). Stability assessment was performed by determining the difference between the averages of the tests determined at the set term and the average of the E1 sample (initial sample, time 0). Using the statistical method of analysis of variability (ANOVA), the repeatability standard deviation was determined, on the basis of which the detection limit of the stability test was established. The detailed calculation is shown in Appendix A. The results of the stability test are shown in Figure 4. The repeatability standard deviations (sr), mean squares between groups MSE_between_, mean squares within groups MSE_within_, s_meas_, and LOD are shown in Table 6 and Table 7, respectively, for TPC and TAC.

The detection limit of the stability test for TPC and TAC was 3.68 mg GAE/100 g and 0.79 mg Cyd-3-Glu/100 g, respectively. The average values of the measured properties are shown in Figure 4. Through polynomial regression, they were established with a degree of fit of R^2^ = 0.9903 for TPC and R^2^ = 0.9864 for TAC from the stability loss versus time curves. The samples kept at in the temperature range between 2 °C and 4 °C for 21 days showed instability, F_TPC_ = 16.63; F_TAC_=24.31 > F_critic_ = 4.07, compared with the initial concentrations. The duration of stability at an accepted level from the initial value can be determined from the regression data. During the total duration of the test, the optimal sample retained 94.8% of the initial TPC and 91.4% of the initial TAC. The stability test indicates that the optimal extract obtained using apple vinegar-acidified solvent has a good short stability.

### 3.4. Red Cabbage Residue Characterization Using Elemental Analysis and ICP-OES

For investigation of the red cabbage residue obtained after pigment recovery, we considered the fact that the current fertilizer Regulations (EU) 2019/1009 [67,68] include a wider range of fertilizing products (inorganic, organo-mineral, and organic fertilizers, soil improvers; liming materials; growing media; inhibitors; plant biostimulants; and fertilizing product blends); and support a greater use of recycled and organic materials as fertilizers for crops.

From the results presented in Table 8, it can be observed that the concentration of contaminants determined by ICP-OES in the red cabbage residue is below the limit values set for an organic fertilizer. The specification for a solid organic fertilizer that contains more than one primary nutrient is also fulfilled. The sum of the nutrient contents was greater than 4% by mass.

The residue obtained after the extraction of anthocyanins and other polyphenols fulfilled the specifications established for the fertilizing products, i.e., corresponding levels of nutrients and contaminants under the threshold limits. This extracted residue is also depleted of anthocyanins and polyphenols with potential allelopathic risks [68]. Therefore, it could be used as an organic fertilizer or in combination with other organic or inorganic fertilizers to return nutrients to the agricultural production system.

## 4. Conclusions

We demonstrated for the first time that apple vinegar can be successfully used for extracting various polyphenols from red cabbage by-products. Using the statistical method for analysis of variance (ANOVA), the standard deviation between red cabbage extract subsamples and the repeatability standard deviation were determined on the basis of which was established the detection limit of the stability test. The study indicates that red cabbage extracts using apple vinegar have a good short stability and high application potential based on their high TPC and TAC. On the basis of the presented results and considering that the contaminants Ni, Cr, Cd, Pb, and Co had values lower than the quantification limits, the obtained extract can be used to improve the color of food and beverage products, being suitable for various formulations in the food or printing industry.

The presented work, which is based on the use of non-toxic solvents, minimizes the organic acid concentration to 1% (*v*/*v*) and the extraction and concentration time to approximately 2 h, together with the possibility of exploiting the residue as an organic fertilizer, and can be considered a sustainable approach for the valorization of red cabbage by-products. This valorization would reduce the pressure of waste disposal on the environment and reduce the use of chemical fertilizers.

Further research is needed to improve the extraction process and to study the scale-up and feasibility of the recovery process of red cabbage by-products taking into account parameters such as safety and shelf-life extension. The possibility of formulating and using the pigments in 3D/4D printing and agrochemical testing of the organic fertilizer will also be evaluated.

## Figures and Tables

**Figure 1 foods-12-04157-f001:**
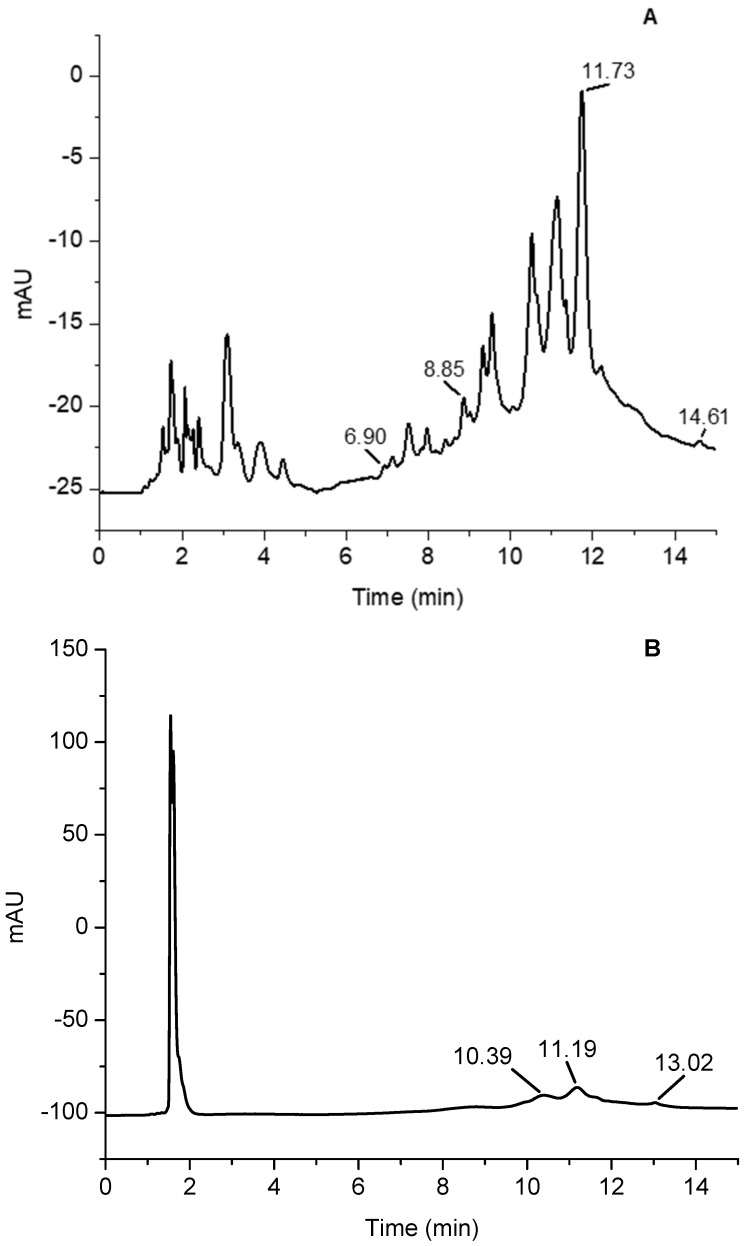
Chromatogram of red cabbage extract (EtOH50%) with apple vinegar (**A**) and without apple vinegar (**B**). Catechin (tr = 6.9 min); Epicatechin (tr = 8.8 min.); Kaempferol (tr = 14.6 min.) and quercetin 3-rutinoside (tr = 11.7 min.), λ = 280 nm.

**Figure 2 foods-12-04157-f002:**
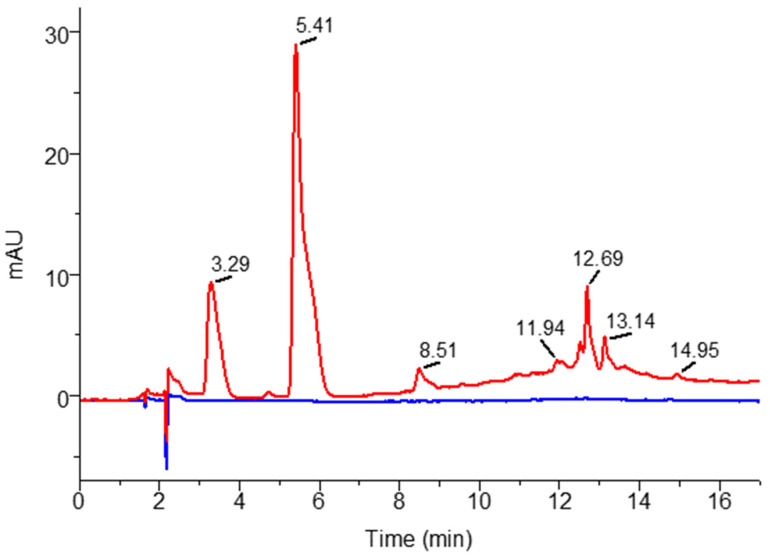
Chromatogram of extraction solvent: EtOH 50% with apple vinegar λ = 280 nm (red line) and 350 nm (blue line).

**Figure 3 foods-12-04157-f003:**
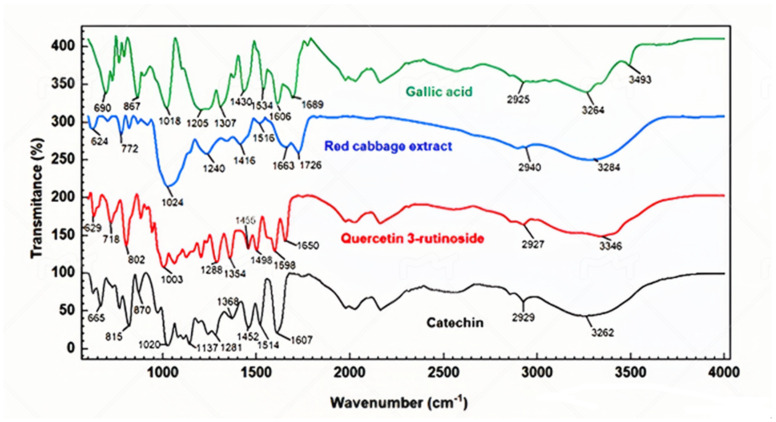
FTIR spectrum of the red cabbage extract obtained by the solvent system EtOH 50% + apple vinegar, compared with the FTIR spectra of gallic acid, quercetin 3-rutinoside, and catechin.

**Figure 4 foods-12-04157-f004:**
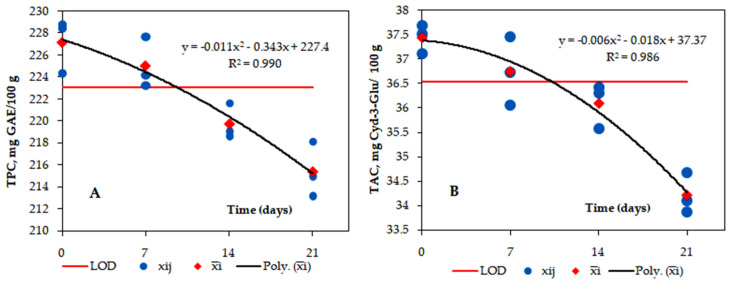
Red cabbage extract—total polyphenol content (TPC) and total monomeric anthocyanin content (TAC) results of the short-term stability in the temperature range between 2 °C and 4 °C (n = 12; a = 4); (**A**)—ANOVA−TPC, EtOH50%−AA1%; (**B**)—ANOVA−TAC, EtOH50%−AA1%.

**Table 1 foods-12-04157-t001:** Summary of the sums of squares and degrees of freedom.

Source of Variation	Sum of Squares	Degrees of Freedom
Between-sample	$n\sum_{i}({\bar{x}}_{i}-\bar{x}{)}^{2}$	a − 1
Within-sample	$\sum_{i}\sum_{j}(x_{ij}-{\bar{x}}_{i}{)}^{2}$	a(n − 1)

*n* = total number of observations, *i* varies from *1* to *n*; a = total number of populations/samples, *j* varies from *1* to *k*; ${\bar{x}}_{i}$ the sample mean for a group; grand mean $\bar{x}$ is the mean of all observations.

**Table 2 foods-12-04157-t002:** Physicochemical composition of the solvent systems before the extraction process.

Solvent System	pH	TPC mg GAE × L^−1^
Ethanol/water/vinegar (30:60:10)	3.04 ± 0.06 cd	18.68 ± 0.61 c
Ethanol/water/vinegar (50:40:10)	2.96 ± 0.06 bc	20.45 ± 0.92 d
Ethanol/water/vinegar (70:20:10)	2.84 ± 0.07 ab	24.18 ± 1.03 e
Bidistiled water	4.43 ± 0.05 f	-
Bidistiled water/vinegar (90:10)	3.17 ± 0.05 d	14.32 ± 0.36 b
Bidistiled water/glacial acetic acid (99:1)	2.71 ± 0.07 a	5.07 ± 0.44 a

Values are mean ± SD (n = 3). TPC—total polyphenols content; GAE—gallic acid equivalent. The pH values were measured with a Consort P901 pH Metter equipped with an electrode HI 1131 (Hanna Instruments) by dilution of solvents at 1:10 (*v*/*v*) with bidistilled water. Mean values followed by the same letter in the same column are not significantly different (*p* ≥ 0.05).

**Table 3 foods-12-04157-t003:** Composition and physicochemical properties of the extracts.

Solvent System	pH	TAC^DW^ mg Cyd-3-Glu/100 g	TPC^DW^ mg GAE/100 g	Color Density	Degradation Index
EtOH 30%-AA1%	3.79 ± 0.07 a	341.29 ± 7.36 c	1589.16 ± 36.40 c	3.63 ± 0.09 c	0.27 ± 0.01 b
EtOH50%-AA1%	3.68 ± 0.06 a	374.70 ± 5.41 d	2272.07 ± 44.51 d	4.12 ± 0.09 d	0.24 ± 0.01 ab
EtOH 70%-AA1%	3.63 ± 0.06 a	367.62 ± 7.36 d	1642.13 ± 40.60 c	3.78 ± 0.10 c	0.25 ± 0.01 ab
Water	4.38 ± 0.07 b	128.57 ± 3.03 a	518.20 ± 19.71 a	1.05 ± 0.04 a	1.05 ± 0.05 c
Water-AA1%	3.69 ± 0.06 a	320.53 ± 7.12 b	1323.00 ± 32.08 b	2.15 ± 0.07 b	0.21 ± 0.01 a

EtOH—ethanol; AA—acetic acid (equivalent % from the apple vinegar); TAC—total anthocyanin concentration; Cyd-3-Glu—cyanidin 3-glucoside; TPC—total polyphenols content; GAE—gallic acid equivalent; DW—dry weight at 105 °C; λmax pH 1 = 526 nm; λmax pH 4.5 = 543 nm for EtOH30, 50, 70%-AA1% and Water-AA1%; λmax pH 4.5 = 521 nm for Water extract. Values are mean ± SD (n = 3). Mean followed by the same letter in the same column are not significantly different (*p* ≥ 0.05).

**Table 4 foods-12-04157-t004:** Results obtained by HPLC-DAD and UV-Vis analysis.

Flavonoids	Domain of the Calibration Curve	R^2^	Concentration ^A^ mg/100 g, DW	Concentration ^B^ mg/100 g, DW
catechin	y = 6.404x + 2.218	0.9994	<1.10 *	<1.10 *
epicatechin	y = 74.575x − 3.128	0.9994	4.46 ± 0.27	<1.10 *
quercetin 3-rutinoside	y = 14.302x − 0.0346	0.9999	62.92 ± 1.52	20.41 ± 0.62
kaempferol	y = 23.459x − 10.979	0.9996	<1.00 *	<1.00 *
total flavonoid content (catechin equivalent)	y = 0.0033x − 0.0012	0.9997	367.73 ± 6.30	205.30 ± 10.33

Results are expressed as mean ± SD (n = 2); * values represent the limit of quantification; DW—dry weight at 105 °C. A = Red cabbage EtOH 50% extract with apple vinegar; B = Red cabbage EtOH 50% extract without apple vinegar.

**Table 5 foods-12-04157-t005:** ICP-OES results of red cabbage extract with 50% EtOH-AA1%.

Element λ (nm)	Domain of the Calibration Curve mg L^−1^/ μg L^−1^	The Correlation Coefficient	Concentration, mg × Kg^−1^, FW (EtOH50%-AA1%)
Ca (317.933)	1–10 mg L^−1^	r = 0.9999	90.10 ± 1.32
Mg (285.213)	1–10 mg L^−1^	r = 0.9999	67.00 ± 0.57
K (766.490)	1–10 mg L^−1^	r = 0.9999	1783.0 ± 10.8
Na (589.592)	1–10 mg L^−1^	r = 0.9998	28.80 ± 1.59
P (213.317)	0.1–1.2 mg L^−1^	r = 0.9976	188.0 ± 18.4
Al (396.153)	0.05–0.5 mg L^−1^	r = 0.9999	<0.6 *
Zn (213.857)	0.05–0.5 mg L^−1^	r = 0.9998	1.23 ± /0.18
Fe (238.213)	0.05–0.6 mg L^−1^	r = 0.9994	3.70 ± 0.76
Mn (257.610)	2–50 μg L^−1^	r = 0.9992	0.51 ± 0.09
Ni (231.604)	2–50 μg L^−1^	r = 0.9997	<0.1 *
Cr (267.716)	2–50 μg L^−1^	r = 0.9994	<0.1 *
Cd (228.802)	2–50 μg L^−1^	r = 0.9994	<0.2 *
Pb (220.353)	5–50 μg L^−1^	r = 0.9971	<0.3 *
Co (228.616)	5–50 μg L^−1^	r = 0.9989	<0.3 *

* Marked values represent the quantification limits of the method; values are mean ± SD (n = 3).

**Table 6 foods-12-04157-t006:** Statistical criteria for performance evaluation of the stability test (ANOVA)-TPC-Red cabbage extract (EtOH50%−AA1%).

Sums of Squares (mg GAE/100 g)		Degree of Freedom		Mean Square Errors (mg GAE/100 g)		Test Criterion, F	Critical Value (95%) F Crit.
SS_Between_	253.18	df _between_	3	MSE_Between_	84.39	16.63	4.07
SS_Within_	40.60	df _within_	8	MSE_Within_	5.07		
s_meas_. (mg GAE/100 g)							1.84
LOD (mg GAE/100 g), P = 95%, k = 2 ^1^							3.68

^1^ k = standard coverage factor; P = level of confidence of approximately 95%; DW—dry weight at 105 °C.

**Table 7 foods-12-04157-t007:** Statistical criteria for performance evaluation of the stability test (ANOVA) −TAC-Red cabbage extract EtOH 50%–AA1%.

Sums of Squares (mg Cyd-3-Glu /100 g)		Degree of Freedom		Mean Square Errors (mg Cyd-3-Glu /100 g)		Test Criterion, F	Critical Value (95%), F Crit.
SS_Between_	17.25	df _between_	3	MSE_Between_	5.75	24.31	4.07
SS_Within_	1.89	df _within_	8	MSE_Within_	0.24		
s_meas_. (mg Cyd-3-Glu/100 g)							0.40
LOD (mg Cyd-3-Glu /100 g), P = 95%, k = 2 ^1^							0.79

^1^ k = standard coverage factor; P = level of confidence of approximately 95%; DW—dry weight at 105 °C.

**Table 8 foods-12-04157-t008:** Red cabbage residue characterization using elemental analysis and ICP-OES.

Parameter	Result, % m/m DW	Specification, DW Regulation (EU) 2019/1009 [67]
C	37.85 ± 0.33	-
H	6.49 ± 0.07	-
N	1.92 ± 0.03	>1%
P_2_O_5_	0.61 ± 0.01	>1% ^a^
K_2_O	1.35 ± 0.001	>1% ^a^
Ca	0.62 ± 0.01	-
Mg	0.091 ± 0.001	-
Cd	* <1.2 mg/Kg	<1.5 mg/Kg
Cr	* <0.2 mg/Kg	-
Cr (VI)	-	<2 mg/Kg
Pb	* <2.3 mg/Kg	<120 mg/Kg
Cu	22.29 ± 1.14	<300 mg/Kg
Zn	31.66 ± 1.23	<800 mg/kg

* Marked values represent the quantification limits of the method. ^a^ Specifications values of 1% for P_2_O_5_ or K_2_O [67].

## Data Availability

All data are contained within the article.

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
