# Peer review of "Sustainable Recovery of Anthocyanins and Other Polyphenols from Red Cabbage Byproducts"

_foods, 2023, doi:10.3390/foods12224157_

Round 1

Reviewer 1 Report

Comments and Suggestions for Authors

The current manuscript entitled "Sustainable recovery of anthocyanins and other polyphenols from red cabbage byproducts" has scientific merit. However, the manuscript needs to be revised. In particular, the methodology must be improved so that the results can be better understood. Solvent selection and extraction procedures need to be rewritten because they are not clear. Improve the methodology for FTIR analysis and review the presentation of results. I also highlight the title and objective presented in the abstract are not consistent with the last sentence presented in the introduction.

Line 83 to 85: the phrase “The short-term stability is a key parameter for further formulation by encapsulation or as pigments in the ink for additive manufacturing, i.e., 3D/4D printing” does not relate to the work presented.

Line 101: the authors must present the characteristics of apple cider vinegar, such as acidity, alcohol content, and pH.

Line 119 to 120: present the pH.

Item 2.3.7: review the writing, “after 7, 14, and 21 days” is repeated.

Table 2: describe the meaning of TPC in the caption. In the methodology, item 2.3.2, it was not described that the phenolic content of the solvents was determined before the extraction process.

Was it expected that the extraction solvent Bidistiled water and the solvent composed of Bidistiled water/glacial acetic acid (99:1) would contain phenolic compounds?

Items 3.1 and 3.2 describing the choice of solvents need to be improved.

Table 4 is not formatted properly.

The authors can consider these points as a way to improve their manuscript substantially. 

Reviewer 2 Report

Comments and Suggestions for Authors

The manuscript presents very interesting results of the study of the content of anthocyanins and other polyphenols in red cabbage extract. The research is generally well designed. The authors investigated the possibility of using apple vinegar for more efficient anthocyanin extraction and came to interesting results. The novelty of this research is adequately highlighted. However, I have some objections, primarily to the way the results were presented.

Some comments are included to improve the manuscript:

Lines 122-123: The sentence is unnecessary.

Line 126: After evaporation, the extract should be dissolved in a known amount of solvent (up to a certain volume), because the results shown in this way are insufficiently precise.

Chapters 2.3.2, 2.3.3, and 2.3.4 are too extensive. It is sufficient to provide appropriate references and only highlight the changes compared to the original procedures.

Table 2: It is necessary to perform a statistical analysis (e.g. Tukey's test)

Table 3: It is not clear which statistical test was used. Authors should explicitly state this information. (Note: It is common to use Tukey's or some similar statistical test to compare samples).

Presentation of results: It would be better (preferable) if the results for TPC, TAC, individual polyphenols... were expressed (converted) to the mass of the raw sample and not to the volume of the extract.

Comparing the results with literature data in this way is not adequate. For example, in ref. [50] TPC results are expressed as mg chlorogenic acid/kg fresh weight.

Lines 279-283: The sentences are too general and well-known. Sentences may be omitted.

Line 295: Reference [52] should not be italicized.

Lines 304-305: The sentence is not clear.

Figure 1: The text needs to be edited; the wavelength should be specified at the end.

Line 319: The reference [54] is to apples and I'm not sure it's relevant to the comparison.

Line 321: Rutin – throughout the text the names of polyphenols should be stated uniformly (either rutin or quercetin-3-rutinoside).

Lines 327-329: The authors present literature data for rutin in mg/L, and in the above references are given as mg/100 g dry weight.

Line 373: y = is specified twice

Line 375: Trolox equivalent [62]. It is necessary to delete the full stop after the reference [62]

Table 5: A space must be inserted after the comma. Otherwise, it's hard to read.

Table 5: It is not clearly indicated in which units the results are expressed.

Line 415: What does the sign “÷” mean in 2÷4°C.

Lines 420-421: Figure 2? It's probably a mistake.

Reviewer 3 Report

Comments and Suggestions for Authors

The manuscript entitled "Sustainable recovery of anthocyanins and other polyphenols from red cabbage byproducts" conforms to the aims and the scope of the Journal.

The aim of this work was to develop a sustainable process for the extraction of anthocyanins from red cabbage by-products by including apple cider vinegar in the extraction solvent. The chemical characterization of this extract was carried out by FTIR, UV-Vis, HPLC-DAD and ICP-OES. The extraction residues are proposed to be used for soil treatment as a fertilization product.

Βelow are suggestions for authors to consider.

> > Introduction Section

Line 70 – 85.  It is recommended to include this part in the discussion.

It is suggested to improve the introduction.

> > Materials and Methods Section

Chemicals, and standards, it is suggested to follow a unified format: company (city, country)

Replace 2.5-Bis(5-tert-butyl-benzoxazol-2-yl)thiophene with 2,5-bis(5-tert-butyl-benzoxazol-2-yl)thiophene,  check the full text.

Line 130-132. Please Correct.

Line 167.  cyd-3-glu:  write in the same way throughout the text.

Line 195. H2O2 30% H2O2 replace with H2O2 30%

Line 198. 0.5-g replace with 0.5g

> >  Results and discussion Section

Please complete (in case of absence):

·       how the results are represented

·       the number of experiments performed.

·       the equations of the calibration curves

·       the correlation coefficient (R²)

Table2.: Solvent system %(v/v): delete: %(v/v)

 TPC. It is recommended to report means and standard deviations to the same decimal place.

Table 3. It is recommended to report means and standard deviations to the same decimal place.

Line 371. It is recommended to report mean and standard deviation to the same decimal place.

Line 373.  Correct y = y=1653.3x+12.989. It also exists on line 178.

Table 5. It is recommended to report means and standard deviations to the same decimal place.

Table 8. Correct Cu: 22.±29±1.14

Table 2 and Figure 4. Please make the letters bold 

Figure 1 and Figure 2 the size of the fonts is too small to read clearly.  Please check the legends. I propose their integration.

>>Additional experiments could be carried out, such as the determination of the Total Flavonoid Content and Total Tannin Content.

> > Further checking of the language and bibliography style is required.

> > Several errors need to be corrected. Some of them:

line 117.  Replace 0C  with °C, check the full text.

Line 27.  Replace ml with mL, check the full text.

Line 84.  Correct 2÷4, check the full text.

Choose the abbreviation you want to use and please follow it throughout the text. Example: Catechin or C?

Reviewer 4 Report

Comments and Suggestions for Authors

The comments are as follows:

1. The Introduction section should be improved by inserting more relevant and new information. Please, exclude well-known facts. More info about red cabbage byproduct. Its production statistics?

2. The novelty of this work should be clearly stated in the Introduction section.

3. Subsection 2.2. More details about UAE are required (device info, sonotrode or bath, ultrasound power/amplitude). 

4. Line 119. The concentration of vinegar in the solvent mixture?

5. Line 130. The best red cabbage extract according to what?

6. Subsection 2.3.2. Please, include calibration curve and correlation coefficient.

7. Future perspective and practical application of the findings should be more highlighted in the conclusions.

8. The text should be checked for typos.
